# Combining Vis-NIR and NIR Spectral Imaging Techniques with Data Fusion for Rapid and Nondestructive Multi-Quality Detection of Cherry Tomatoes

**DOI:** 10.3390/foods12193621

**Published:** 2023-09-28

**Authors:** Fei Tan, Xiaoming Mo, Shiwei Ruan, Tianying Yan, Peng Xing, Pan Gao, Wei Xu, Weixin Ye, Yongquan Li, Xiuwen Gao, Tianxiang Liu

**Affiliations:** 1College of Mechanical and Electrical Engineering, Shihezi University, Shihezi 832003, China; tfnszbd@163.com (F.T.); mxmshz@126.com (X.M.); 2College of Information Science and Technology, Shihezi University, Shihezi 832003, China; sound_euphonium1997@outlook.com (S.R.); pengxing20200512@163.com (P.X.); yeweixin@stu.shzu.edu.cn (W.Y.); 18061763059@163.com (Y.L.); 17791828853@163.com (X.G.); 3Key Laboratory of Northwest Agricultural Equipment, Ministry of Agriculture and Rural Affairs, Shihezi 832000, China; 4Engineering Research Center for Production Mechanization of Oasis Characteristic Cash Crop, Ministry of Education, Shihezi 832000, China; 5School of Electronic Information and Electrical Engineering, Shanghai Jiao Tong University, Shanghai 201100, China; yantianying@163.com; 6College of Agriculture, Shihezi University, Shihezi 832003, China; liutianxiang0127@163.com

**Keywords:** cherry tomato, hyperspectral, data fusion, quality inspection

## Abstract

Firmness, soluble solid content (SSC) and titratable acidity (TA) are characteristic substances for evaluating the quality of cherry tomatoes. In this paper, a hyper spectral imaging (HSI) system using visible/near-infrared (Vis-NIR) and near-infrared (NIR) was proposed to detect the key qualities of cherry tomatoes. The effects of individual spectral information and fused spectral information in the detection of different qualities were compared for firmness, SSC and TA of cherry tomatoes. Data layer fusion combined with multiple machine learning methods including principal component regression (PCR), partial least squares regression (PLSR), support vector regression (SVR) and back propagation neural network (BP) is used for model training. The results show that for firmness, SSC and TA, the determination coefficient R^2^ of the multi-quality prediction model established by Vis-NIR spectra is higher than that of NIR spectra. The R^2^ of the best model obtained by SSC and TA fusion band is greater than 0.9, and that of the best model obtained by the firmness fusion band is greater than 0.85. It is better to use the spectral bands after information fusion for nondestructive quality detection of cherry tomatoes. This study shows that hyperspectral imaging technology can be used for the nondestructive detection of multiple qualities of cherry tomatoes, and the method based on the fusion of two spectra has a better prediction effect for the rapid detection of multiple qualities of cherry tomatoes compared with a single spectrum. This study can provide certain technical support for the rapid nondestructive detection of multiple qualities in other melons and fruits.

## 1. Introduction

Cherry tomato is a popular respiratory leaping fruit with bright color, hard fruit, sweet and sour taste, and is rich in sugar, organic acids and other nutrients [1]. Its internal quality is an important basis for measuring its nutritional value. In general, cherry tomatoes are mainly evaluated by testing the soluble solid content (SSC) and titratable acidity (TA), firmness and other indicators [2]. Acidity and sugar affect their taste comfort and can be used for grading internal quality; changes in firmness can be used for quality monitoring during storage and transportation of cherry tomatoes and for determining the ripeness of the fruit [3]. Traditionally, the SSC and TA of cherry tomatoes are tested by the refractometer method, which requires the fruit to be extruded into juice and then titrated, and the firmness is tested with the fruit firmness tester. However, the above methods have a single index, are time-consuming and destructive and are difficult to be applied to the batch testing of fruits [4].

Spectral analysis is a widely used nondestructive analysis method for fruit inspection. Feng et al. obtained the spectra of cherry tomatoes using a portable near-infrared spectrometer (950–1650 nm) and applied partial least squares (PLS) and support vector machines (SVMs) for nondestructive prediction of postharvest firmness, SSC, and pH of cherry tomatoes [5]. Borba et al. analyzed internal quality indicators such as SSC and TA of different types of fresh tomatoes using standard measurements and correlated these reference values with spectral data using the partial least squares regression (PLSR) to establish a nondestructive testing model [6]. However, visible and near-infrared spectral detection techniques are susceptible to limited spectral bands and the spectral information obtained is not comprehensive [7]. Hyperspectral imaging technology can detect two-dimensional spatial information and one-dimensional spectral information at the same time and thus combining image and spectral features, through which it can get the overall spatial spectral information of the fruit [8]. Shao et al. used visible and near-infrared (Vis-NIR) hyperspectral imaging for rapid detection and visualization of SSC of two sweet potato species, determined the optimal spectral preprocessing and established multiple prediction models for sweet potato SSC using the partial least squares regression (PLSR), support vector regression (SVR) and multiple linear regression (MLR), and comparatively analyzed the models with the best prediction performance [9]. Li et al., (2022) utilized the short-wave infrared (SWIR) hyperspectral imaging system (HIS) in the spectral range of 1000–2500 nm in conjunction with multivariate regression modeling to predict the SSC of dried Hami jujube [10]. Ye et al. used a Vis-NIR (376–1044 nm) and near-infrared (NIR) (915–1699 nm) HIS combined with a machine learning algorithm to efficiently detect pesticide residue levels in grapes, respectively [11]. Xiang et al., (2022) carried out nondestructive testing of SSC and firmness of cherry tomatoes based on Vis-NIR hyperspectral images and multiple regression prediction models, and the results of the study showed that hyperspectral imaging technology has the potential to be applied in the prediction of the quality of cherry tomatoes, which provides a new option for nondestructive testing of the quality of cherry tomato fruits [12]. Zhao et al., (2023) used NIR hyperspectral imaging technology (980–1660 nm) for the detection of firmness, SSC, lycopene and TA content of processing tomatoes and to classify fruits at three stages of ripening [13].

Most of the current studies are using a single band of spectral region for detection; however, the spectral response of different bands is not the same. The fusion of multiple data information can better interpret the detected object than single data information, and complementary information can be analyzed through the fusion of information, which can provide a complete description of the object to be detected [14]. Orlandi et al., (2019) utilized complementary information obtained from electronic nose and electronic tongue sensing systems using a data layer fusion strategy to rapidly quantify grape ripening [15]. Ding et al., (2021) integrated watermelon tapping vibration signals with Vis-NIR spectral data, which were subjected to standard normal transform (SNV), to generate a photoacoustic fusion curve [16]. This approach enabled the reflection of a more comprehensive range of watermelon firmness information. Hu et al., (2023) examined different types of pesticide residues on the surface of cantaloupe melon using VIS-NIR and SWIR hyperspectral imaging systems for four pesticides commonly used in cantaloupe melon, and compared the effectiveness of single-band spectral ranges and information fusion in the classification of different pesticides. The results showed that the use of the spectral ranges of the fused information to categorize the pesticide residues was more effective [17]. Li et al., (2023) combined two hyperspectral imaging techniques (Vis-NIR and NIR) with a data layer fusion strategy for the prediction of multiple quality metrics of chicken meat under different storage conditions, and used partial least squares regression (PLSR) to establish quantitative predictions after preprocessing, which showed that the fused spectra exhibited better performance on multiple quality metrics compared to individual spectral information [18].

Therefore, given that the spectral information of the two bands may complement each other, this paper will explore the fusion of the spectral information of the two bands and the combination of machine learning algorithms to establish a multi-quality prediction model for SSC, TA and firmness of cherry tomatoes, which will provide technical support for the quality grading and rational storage and processing of cherry tomatoes.

## 2. Materials and Methods

### 2.1. Plant Materials

The fresh cherry tomatoes utilized in the experiment were procured from Oasis Jiuding Farmer’s Market in Shihezi City. A total of 200 cherry tomatoes with consistent ripeness, uniform size, devoid of mechanical damage and without any signs of pests or diseases were carefully chosen. These selected tomatoes were then placed under experimental conditions of 25 °C temperature and 50% relative humidity for a duration of 12 h. The samples were cleaned and numbered one by one prior to the experiment. The spectral image data of the cherry tomatoes were completed within one day. SSC, TA and firmness data were measured.

### 2.2. Hyperspectral Imaging Acquisition and Preprocessing

In this study, two hyperspectral imaging systems (Vis/NIR and NIR hyperspectral imaging systems) were used to photograph cherry tomato samples. Both the Vis/NIR and NIR hyperspectral imaging systems consist of four modules, including an imaging module, an illumination module, a lifting module and a software module, as shown in Figure 1. The imaging module includes the Surface Optics Corporation (SOC) 710 series camera (Surface Optics Corporation, San Diego, CA, USA). These cameras have an internal scanning mechanism to be able to scan in any direction or directly vertically downward without the need for an additional scanning stage. The Vis/NIR hyperspectral imaging system (SOC 710VP) has a spectral wavelength range of 376–1044 nm, a spectral resolution of 5 nm, and 128 bands. The NIR hyperspectral imaging system (SOC 710SWIR) has a spectral wavelength range of 915–1699 nm, spectral resolution of 2.7 nm, and 288 bands. A halogen lamp was used as the illumination module, and the power of a single halogen lamp was 50 W. A total of four halogen lamps were used. The platform module placed the subject, the imaging module allowed us to complete the capture of the subject, and the software module was used to control the acquisition of the HSI. Due to the convex surface of the samples, the uneven reflection creates a highlighted region near the vertical axial. Thus, we used ENVI5.3 (ITT, Visual Information Solutions, Boulder, CO, USA) to avoid the highlight region and to extract the reflection value for each band from the region of interest.

The original hyperspectral image is corrected to a reflectance image using a grayscale reference image. The correction is performed by the following equation:I_r_ = (I_raw_ − I_dark_)/(I_white_ − I_dark_)(1)
where I_r_ is the reflectance image, I_raw_ is the original image, I_white_ is the all-white reference image and I_dark_ is the all-black reference image. The grayscale reference image consists of 50% I_dark_ and 50% I_white_.

In order to better extract the information from the signal, this noise needs to be removed and therefore the spectra need to be preprocessed. Preprocessing the spectral data helps to improve the regression model performance. In this study, the Savitzky Golay (SG) [19] smoothing filter (polynomial order 0, kernel size 3) was used to improve the smoothness of the original average spectral data and reduce noise interference. Then, standard normal variable transformation (SNV) was used to avoid the influence of surface scattering, solid particle size and optical path changes in diffuse reflectance spectra [20]. Spectral data analysis processing was performed using ENVI5.3 and spectral preprocessing was performed in Unscrambler × 10.4 (CAMO, Oslo, Norway).

### 2.3. Measurement of Physical and Chemical Indicators

After spectral acquisition, the firmness parameters of cherry tomatoes were measured with a TA.XT plus texture analyzer (Stable Micro Systems, Inc., London, UK) as shown in Figure 2. The test was performed using a P50 probe descending at 1 mm/s before contacting the specimen, and compressing it at 1 mm/s after contact. The specimen was compressed by 8 mm, and then the probe returned to the origin of the test at a speed of 1 mm/s. After the compression was completed, the cherry tomatoes were sliced and squeezed, and the soluble solid content (SSC) was determined with a hand-held digital brix refractometer (PR-101α, Atago Co., Ltd., Tokyo, Japan). Titratable acid was measured using an acidometer (PR-101α, Atago Co., Ltd., Tokyo, Japan). SSC and TA were measured three times and averaged [21].

### 2.4. Modeling Algorithm

#### 2.4.1. Principal Component Regression

PCR is a multivariate statistical analysis method that utilizes the idea of dimensionality reduction to convert multiple indicators into a few comprehensive indicators with minimal information loss. The comprehensive indicators generated by transformation are usually referred to as principal components (PCs), where each principal component is a linear combination of the original variables and is not related to each other, resulting in superior performance of the principal components compared to the original variables [22]. Generally, the principal component variables were selected based on whether the cumulative contribution reached 85%, and the number of principal component factors was selected as 5 in this study.

#### 2.4.2. Partial Least Squares Regression

PLSR is the most commonly used modeling method in chemometrics to analyze the correlation between spectral data and reference quality indicators, which is widely applicable in fruit and vegetable quality studies [23]. PLSR is a multivariate statistical analysis method that projects the independent variables and dependent variables into a new multidimensional space, constitutes the prediction matrix X and the response matrix Y and decomposes them to extract the principal factors. Then, it arranges these factors according to the correlation between them from the largest to the smallest correlation, and establishes a linear regression model. In this paper, the number of PLSR factors is selected as 7, according to the correlation size of the principal factors.

#### 2.4.3. Support Vector Regression

SVR is a supervised machine learning algorithm based on support vector machines that models linear and nonlinear regression prediction of datasets. SVR maps low-dimensional space vectors to high-dimensional space by introducing a kernel function and constructs a linear decision function in the high-dimensional space, which realizes a nonlinear decision in the original space. To a certain extent, SVR algorithms avoid dimensional catastrophe and overfitting. Radial basis function (RBF) was chosen as the kernel function [24]. The RBF hyperparameters, including the penalty coefficients (c) and kernel function parameters (g), were optimized using the grid search (GS) algorithm and 10-fold cross-validation method.

#### 2.4.4. Back Propagation Neural Network

BP is a multi-layer feed-forward neural network trained using error back propagation algorithm and is one of the most widely used neural network models. The neural network algorithm is mainly characterized by forward propagation of the signal and backward propagation of the error. If the output layer does not get the desired output, the weights and thresholds are adjusted according to the prediction error until the predicted value is close to the target value. Due to the good robustness and adaptive ability of this network algorithm, it has been widely used in fruit internal quality detection [25]. The construction of the model using BPNN is mainly divided into two steps. The first step is the training process of the model, which uses the training set to train the model. The second step is the testing process, which uses the test set to test the model and examine the accuracy of the model prediction results. The structure of the BP neural network model mainly consists of an input layer, a hidden layer and an output layer. Both the input and output layers are single-layered, while a single hidden layer is utilized. The number of neurons in the hidden layer is set to 10. In this study, the tansig activation function was used for the hidden layer activation function, and trainlm was used as the training function, with the number of iteration steps set to 6 by default; the learning rate was set to 0.01; and the training interval was 25.

#### 2.4.5. Data Fusion

Information fusion technology has the advantage of more information and better fault tolerance than single detection technology. Information fusion can realize the process of fusion of multiple sources of information at multiple levels. According to the level of data abstraction in the fusion system, fusion can be categorized into data layer fusion, feature layer fusion and decision layer fusion.

Data layer fusion refers to the direct correlation of the raw data from each sensor, which is sent to the fusion center to complete the comprehensive evaluation of the measured object. Feature layer fusion means that the original data is processed through feature extraction, association and normalization, and then sent to the fusion center for analysis and synthesis to complete the comprehensive evaluation of the object that is being tested. Decision level fusion refers to the localization of the signals from each sensor prior to fusion. Feature layer fusion methods may be more suitable for fusing two different types of sensor data. For data layer fusion, which requires the need for parallel sensors, the data layer fusion method has minimal information loss and higher fusion accuracy.

This paper discusses the fusion of raw data layer data, which only requires the direct concatenation and merging of data from samples of different testing instruments. Based on the fused data, the physical and chemical properties of the samples can be analyzed to obtain more analytical results than a single spectroscopic technique. In this study, 354 variables were generated by merging data from Vis-NIR and NIR spectroscopic instruments, which contributed 116 and 238 variables, respectively [26].

### 2.5. Model Evaluation

To evaluate the predictive ability of the model, the following coefficients of determination were used: the calibration and prediction set (R^2^_C_ and R^2^_P_), the root mean square error for the calibration and prediction set (RMSEC and RMSEP). In general, good models have higher values of R^2^_C_ and R^2^_P_ and lower values of RMSEC and RMSEP [27].
(2)$$R^{2}=1-\frac{\sum_{i}{\left({\hat{y}}_{i}-y_{i}\right)}^{2}}{\sum_{i}{\left(\bar{y}-y_{i}\right)}^{2}}$$
(3)$$RMSE=\sqrt{\frac{1}{m}\sum_{i=1}^{m}{\left(y_{i}-{\hat{y}}_{i}\right)}^{2}}$$
where m represents the number of samples, ${\hat{y}}_{i}$ represents the predicted value, $y_{i}$ represents the actual value and $\bar{y}$ represents the mean value of the actual value.

## 3. Results and Discussion

### 3.1. Reference Measurements

Reference measurements of firmness, SSC and TA of cherry tomato fruits were performed according to the destructive method described in the “Reference measurements of firmness and SSC, TA” section. The range, mean and standard deviation values of firmness and SSC and TA of the fruits were analyzed and presented in Table 1. The SSC of 200 cherry tomato samples varied from 6.10% to 8.95%, with a mean of 8.03% and a standard deviation of 0.61%; the TA varied from 3.10% to 3.98%, with a mean of 3.60% and a standard deviation of 0.21%; and firmness varied from 14.21 N to 21.97 N, with a mean value of 18.51 N and a standard deviation of 1.70 N. This relatively high variability may be attributed to the fact that cherry fruits were harvested from different plants and at different growth locations.

For calibration modeling purposes, all cherry samples were randomly divided into a calibration set of 150 samples and a prediction set of 50 samples, and the statistics for firmness, SSC and TA are shown in the last 2 rows of Table 1. The firmness in the calibration set ranged from 14.53 N to 21.46 N, and the firmness in the prediction set ranged from 14.21 N to 21.97 N; the corresponding SSC ranged from 6.25–8.60% and 6.10–8.95%; and the TA ranged from 3.22–3.92% and 3.10–3.98%. The relatively wide range of data variability in calibration and prediction of centralized firmness, SSC and TA during the modeling process may contribute to a robust calibration model.

### 3.2. Spectral Analysis and Preprocessing

The average spectra and reflectance of cherry tomato samples in the Vis-NIR and NIR are shown in Figure 3. Figure 3a shows the reflectance spectra of 200 cherry tomato samples at 400–1004 nm. Figure 3b shows the reflectance spectra of cherry tomato samples at 1002–1652 nm. It can be seen that the variation trend of the spectral curves of cherry tomatoes with different quality parameters is similar, but the reflection intensity is different, indicating that the types of substances inside the fruits of different cherry tomatoes are similar, but the content is different. This theoretically verifies the feasibility of using hyperspectral technology to detect the internal quality parameters of cherry tomatoes.

The reflectance spectra were then processed using the standard normalized variable (SNV), as seen in Figure 4, which removes the effects of solid particle size and optical path variations on the spectra, and also corrects for spectral errors caused by sample scattering. The spectral trend of each sample is similar due to the same reflective substance. Carotenoids are present in ripe tomatoes and cherry tomatoes have strong absorption bands at 750–850 nm and 1000–1200 nm.

It can be observed that there are clear spectral absorption valleys at 580–590 nm, 650–700 nm, 950–1000 nm, 1150–1200 nm and 1390–1450 nm, as shown in Figure 4. These absorption valleys may be related to the periodic stretching vibrations of C-H, O-H and N-H bonds, which are the most fundamental bonds in organic compounds. In addition, O-H and C-H bonds are related to SSC, TA and water, and the water content affects the cell turgor pressure of the flesh cells, which in turn affects the firmness of the flesh. Therefore, with proper processing, changes in the spectral curve can reveal certain hidden information, such as SSC, TA and firmness.

### 3.3. Data Fusion and Model Construction

The spectral curves are fused with data layers and normalized. Normalization (Nor) removes the effect of different scales between the data, removes the noise information, and allows comparisons and calculations on different scales or units. It can be seen that the obvious two peaks are still at 750–850 nm and 1000–1200 nm, as shown in Figure 5. Quality detection models for predicting SSC, TA and firmness were developed using three different bands in the wavelength range of 400–1004 nm, 1002–1652 nm and 400–1652 nm using PCR, PLSR, SVR and BP yanking different modeling methods. Prior to modeling, values that were clearly anomalous in the experimental works were eliminated. The relationship between the measured values of SSC, TA and firmness after normalized spectral processing and the predicted values of the four models are shown in Table 2.

For the SSC of cherry tomatoes, it can be seen from Table 2 and Figure 6 that under the single spectral band and fused spectral band, compared to PCR, PLSR and SVR prediction models, the BP model predicted a higher set of R^2^ values, smaller RMSE values, higher prediction accuracy of the model and a smaller range of error. The measured and predicted values of the SSC of cherry tomato in the calibration and prediction sets of the BP model are on both sides of the 45° line, with the smallest degree of dispersion, indicating that the model has the best fitting accuracy and good model stability. The SVR model prediction accuracy and range of error were close to the prediction performance of the BP mode, and the PLSR and PCR models. The prediction performance of the PLSR and PCR models is much lower than that of the BP and SVR models, and the model prediction accuracy of the PCR is the lowest but has the largest prediction bias. This may be due to the good learning and prediction abilities of BP neural networks and SVR used for regression in nonlinear regression. Among them, the prediction performance of different prediction models in the fusion bands are all improved compared to the individual bands, in which the R^2^ values of the BP model and SVR model in the fusion bands are both greater than 0.9, which indicates that the fusion band spectral features provide richer information about the internal qualities, as shown in the SSC fusion spectra (Figure 6). Among the three bands, the accuracy of the NIR band prediction model is relatively low, which may be due to the low correlation between the NIR band spectral features and the SSC metrics.

For the TA of cherry tomatoes, it can be seen from Table 2 that the prediction performance of the BP model was still better than the other three models in individual spectral bands and fusion spectral bands. The prediction performance of the BP and SVR models was still better than that of the PLSR and PCR models, and the model prediction accuracy of the PLSR was the lowest in terms of R^2^ and the largest in terms of root mean square error, as shown in the TA fusion spectra (Figure 7). The distance between the correction set and prediction set fitting lines of the PLSR model is relatively large, and the degree of data point dispersion is high, indicating that the PLSR model has poor prediction fitting accuracy and model stability for TA quality. The prediction performance of the fusion band spectral features is still higher than that of the individual band spectral features. Among the three bands, the prediction accuracy of the fusion band BP prediction model reaches 0.922 and the accuracy of the NIR band prediction model is relatively low, which may be due to the low correlation between the NIR band spectral features and the SSC metrics.

For the firmness of cherry tomato, it can be seen from Table 2 that the SVR model prediction set in the Vis-NIR band has an R^2^ value of 0.739 and an RMSE value of 0.862, which resulted in a higher prediction accuracy and a smaller error range compared to the PCR, PLSR and BP models. In the NIR band, the SVR model prediction set has an R^2^ value of 0.692 and an RMSE value of 0.861. The prediction accuracy of the model in the NIR band is slightly lower compared to the Vis-NIR band, which may be due to the lower correlation between the spectral features and firmness metrics in the NIR band. The prediction accuracy of the model in the fusion band is significantly improved, and the prediction performance of the SVR model is better than that of the PLSR, PCR and BP models, in which the prediction performance of the PCR and PLSR models is poorer. Furthermore, the prediction performance of the BP and SVR models is comparatively better, as shown in the firmness fusion spectrum shown in Figure 8, which may be due to the existence of more nonlinear relationships between spectral features and the internal quality indexes, and due to the nonlinear models of the BP and SVR being more effective for the spectral feature information. In the future, deep learning feature extraction can be further carried out on spectral information to explore more correlations between spectral information and firmness quality in order to improve the performance and stability of firmness quality prediction models in cherry tomatoes.

In terms of overall model prediction performance, the prediction model built by fusing spectral features of bands obtained better prediction performance than the prediction model of single spectral features. Among the three indexes of SSC, TA and firmness, spectral features had the best prediction performance for SSC, followed by TA, and firmness was relatively the lowest, which may be due to the fact that the internal composition of fruits and vegetables of SSC and TA have a stronger correlation with the spectral information, and the firmness belongs to the fruit and vegetables of which the viscoelastic property has a weaker correlation with the spectral information. The viscoelastic properties of the fruit and vegetable have a weaker correlation with the spectral information, but the spectral technique still shows high prediction accuracy for multiple qualities of cherry tomatoes, especially the model established by fusion spectroscopy that provides a wider range of spectral band information, which has a greater potential for application.

### 3.4. Visual Analysis of Wavelength Contribution of the Optimal Model

According to the above findings, for SSC and TA, the BP-based prediction model shows better prediction results; for firmness, the prediction model based on SVR shows better results. For the optimal model, visual analysis of wavelength contribution is used to analyze the importance based on the test set, as shown in Figure 9. For SSC (Figure 9a), in the Vis-NIR spectral band, the wavelengths in the 550–600 nm bands all have a high contribution to the SSC metrics; in the NIR spectral band, the wavelengths in the 1450–1500 nm band have a higher contribution to the SSC metrics. The wavelengths of both the Vis-NIR and NIR spectral bands reflect the changes in the SSC metrics, but the Vis-NIR band exhibits a higher distribution of contributions, which may explain that the results of the prediction models built in the Vis-NIR band are better than those built in the NIR band. For TA (Figure 9b), the wavelengths in the bands of 450–550 nm, 650–700 nm and 900–1000 contributed more to the Vis-NIR spectral band; in the NIR spectral band, the wavelengths in the bands of 1350–1450 nm and 1600–1650 nm also contributed more to the TA. Both Vis-NIR and NIR provided different wavelength band information of TA quality in cherry tomatoes, which may explain the improvement in the predictive performance of the TA fusion spectra over the results of single spectral prediction performance. For firmness (Figure 9c), wavelengths in the range of 450–500 nm and 550–600 nm have large contribution values and are mostly higher than 0.010 in the Vis-NIR spectral band; in the NIR spectral band, the wavelength contribution values are mostly less than 0.005, but the overall spectral contribution of firmness is not high relative to SSC and TA—which is more consistent with the results that the model prediction performance in the Vis-NIR spectral band is better than that of the model prediction performance in the NIR band, and has a lower firmness prediction performance. Vis-NIR spectral band model prediction performance is better than the NIR band model prediction performance but the firmness prediction performance is lower, which is more consistent with the results.

## 4. Conclusions

In this study, two hyperspectral imaging techniques with different spectral ranges and information fusion methods were used to enable accurate nondestructive detection of multiple qualities of cherry tomatoes. The prediction performance of several machine learning models for nondestructive detection of different qualities was compared. Considering that the two spectral regions have complementary information, the spectral information of the two strips was fused in a data layer to establish a model for the nondestructive detection of multiple qualities of cherry tomatoes. The results show that the detection results of the Vis-NIR spectra are better than the NIR spectra. However, the information from the fusion of the two spectra provided better detection than the information from the individual spectra. For firmness, SSC and TA, the determination coefficient R^2^ of the multi-quality prediction model established with the Vis-NIR spectra is higher than that of the NIR spectra. The R^2^ of the optimal model obtained by using the BP in SSC and TA fusion bands was greater than 0.9, which is several percent higher than that of a single spectrum. The R^2^ of the best model obtained with SVR in the firmness fusion band was greater than 0.85, which is several percent higher than that of a single spectrum. The prediction results of the model may be related to the wavelength contribution of different spectral bands to the three predicted qualities. To summarize, hyperspectral imaging can be used as a nondestructive method for detecting multiple qualities of cherry tomatoes. The fusion of spectral information can be utilized to effectively improve the prediction accuracy of cherry tomato quality detection. In addition, this method can also provide a reference for the multi-quality detection of other fruits.

## Figures and Tables

**Figure 1 foods-12-03621-f001:**
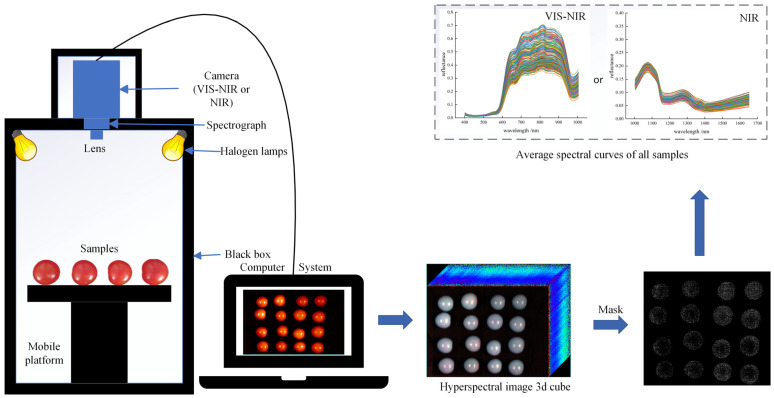
A hyperspectral imaging flow for obtaining the average spectrum of cherry tomatoes.

**Figure 2 foods-12-03621-f002:**
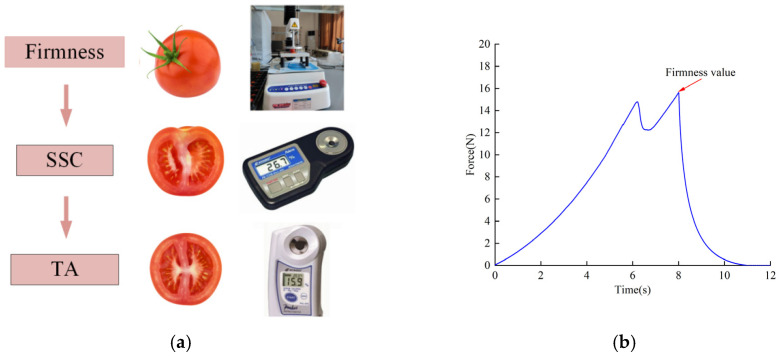
Cherry tomato multi-quality index destructive measuring instrument. (**a**) Multi-quality index destructive measurement process. (**b**) Firmness compression curve.

**Figure 3 foods-12-03621-f003:**
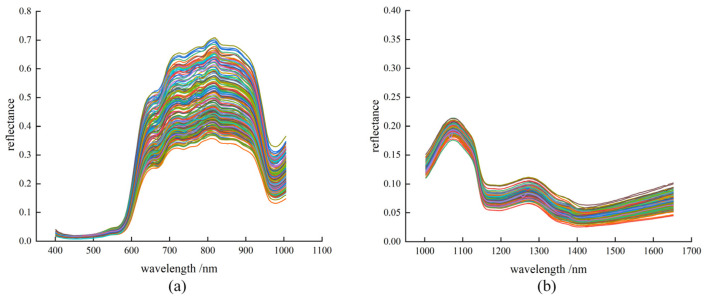
Raw average spectra of cherry tomatoes. (**a**) Vis-NIR raw average spectra. (**b**) NIR raw average spectra.

**Figure 4 foods-12-03621-f004:**
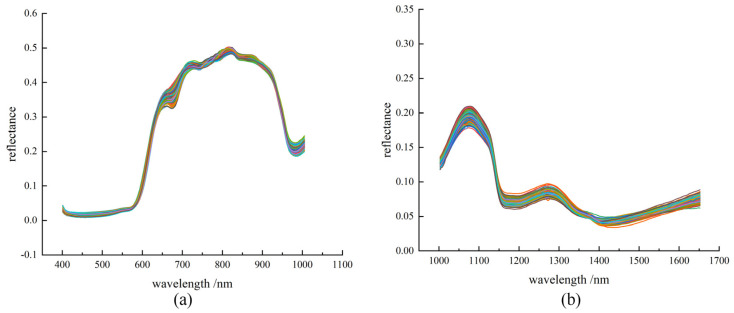
Spectra after SNV pretreatment of cherry tomatoes. (**a**) Vis-NIR spectra after SNV pretreatment. (**b**) NIR spectra after SNV pretreatment.

**Figure 5 foods-12-03621-f005:**
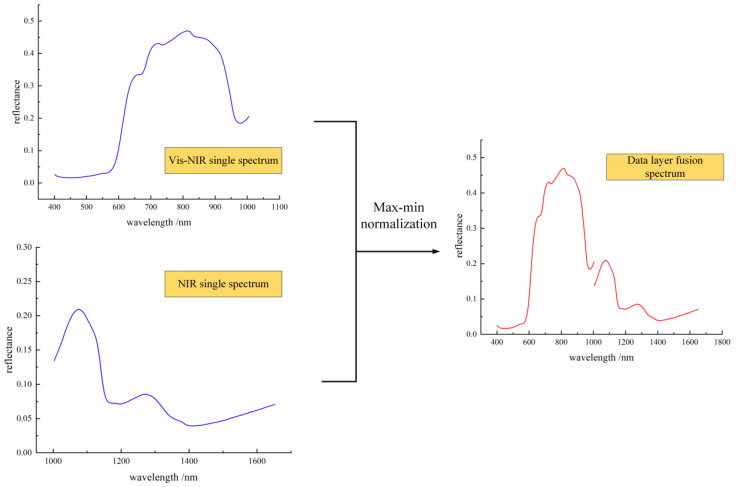
Fusion process of hyperspectral data layers in different bands.

**Figure 6 foods-12-03621-f006:**
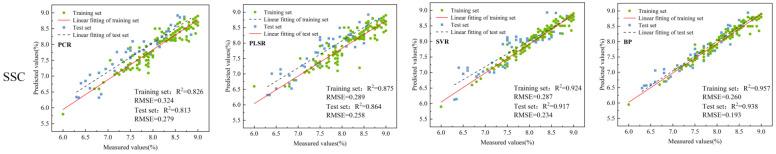
Analysis of SSC prediction results based on hyperspectral data fusion of cherry tomatoes.

**Figure 7 foods-12-03621-f007:**
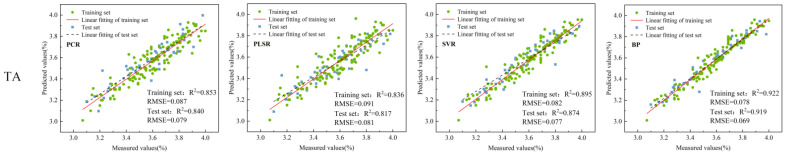
Analysis of TA prediction results based on hyperspectral data fusion of cherry tomatoes.

**Figure 8 foods-12-03621-f008:**
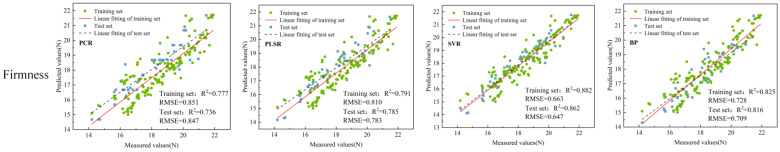
Analysis of firmness prediction results based on hyperspectral data fusion of cherry tomatoes.

**Figure 9 foods-12-03621-f009:**
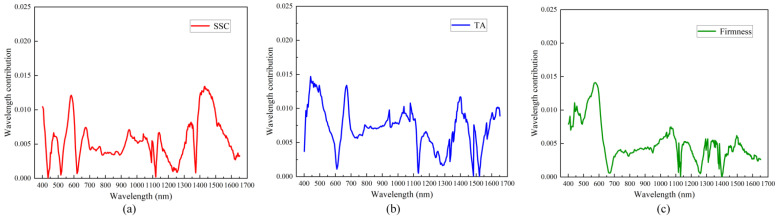
Visualization of the wavelength contribution for different qualities. (**a**) BP model for SSC; (**b**) BP model for TA; (**c**) SVR model for firmness.

**Table 1 foods-12-03621-t001:** SSC, TA and firmness reference measurements in cherry tomatoes.

Sample Set	Number of Samples	SSC (%)			TA (%)			Firmness (N)		
		Range	Mean	SD	Range	Mean	SD	Range	Mean	SD
All samples	200	6.10–8.95	8.03	0.61	3.10–3.98	3.60	0.21	14.21–21.97	18.51	1.70
Calibration set	150	6.25–8.60	8.15	0.53	3.22–3.92	3.62	0.20	14.53–21.46	18.66	1.64
Prediction set	50	6.10–8.95	7.68	0.69	3.10–3.98	3.57	0.23	14.21–21.97	18.07	1.81

**Table 2 foods-12-03621-t002:** Performance parameters for multi-quality prediction of cherry tomatoes for PCR, PLSR, SVR and BP models under different spectral types.

Spectral Type	Model	SSC (%)				TA (%)				Firmness (N)			
		R^2^_C_	RMSEC	R^2^_P_	RMSEP	R^2^_C_	RMSEC	R^2^_P_	RMSEP	R^2^_C_	RMSEC	R^2^_P_	RMSEP
Vis-NIR (400–1004 nm)	PCR	0.757	0.351	0.741	0.328	0.737	0.121	0.757	0.099	0.665	0.977	0.662	0.981
	PLSR	0.763	0.348	0.749	0.299	0.713	0.124	0.718	0.101	0.679	0.953	0.677	0.94
	SVR	0.793	0.331	0.792	0.306	0.766	0.108	0.791	0.09	0.748	0.886	0.739	0.862
	BP	0.850	0.290	0.837	0.267	0.808	0.095	0.822	0.088	0.695	0.903	0.698	0.88
NIR (1002–1652 nm)	PCR	0.637	0.397	0.622	0.370	0.646	0.125	0.636	0.114	0.576	0.967	0.58	0.94
	PLSR	0.709	0.412	0.704	0.316	0.61	0.138	0.604	0.124	0.598	0.899	0.608	0.892
	SVR	0.745	0.316	0.736	0.309	0.693	0.140	0.704	0.118	0.696	0.863	0.692	0.861
	BP	0.805	0.305	0.792	0.289	0.77	0.119	0.758	0.105	0.617	0.886	0.621	0.88
Fusion spectrum (400–1652 nm)	PCR	0.826	0.324	0.813	0.279	0.853	0.087	0.84	0.079	0.777	0.851	0.736	0.847
	PLSR	0.875	0.289	0.864	0.258	0.836	0.091	0.817	0.081	0.791	0.81	0.785	0.783
	SVR	0.924	0.287	0.917	0.234	0.895	0.082	0.874	0.077	0.882	0.663	0.862	0.647
	BP	0.957	0.260	0.938	0.193	0.922	0.078	0.919	0.069	0.825	0.728	0.816	0.709

## Data Availability

All relevant data presented in this article are kept at the request of the institution and are therefore not available online. However, all data used in this manuscript are available from the corresponding authors.

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
