# Peer review of "Combining Vis-NIR and NIR Spectral Imaging Techniques with Data Fusion for Rapid and Nondestructive Multi-Quality Detection of Cherry Tomatoes"

_foods, 2023, doi:10.3390/foods12193621_

Round 1
Reviewer 1 Report
While authors obtain good results, they do not explain why. What is NIR sensitive to? Why is NIR able to predict SSC, TA and firmness? There is no discussion of specificity, just accuracy. Authors just took some spectra, applied a number of algorithms and got some results they deemed acceptable. While it may be useful, they need to explain why they models work. Otherwise, while the article may have practical value, it has very limited scientific value.
SSC and TA are defined on line 51 but are first used on line 18 (abstract) and 39 (introduction). These terms should be defined in both the abstract and the introduction.
Line 136 – a typical equation would apply a -log 10 to equation 1 to obtain absorbance spectra. The -log10 transformation help linearize the data. Why was it not applied here?
Figure 1 shows clear lighting issues where some tomatoes are exposed to much more light than others. How was this addressed in the data? Tomatoes are not flat and so the spectra of pixels on the edge of tomatoes will be significantly different from the spectra of pixels on the top. How was that managed?
The structure of the ANN should be described. The number of PCR and PLS factors should be provided.
Authors seem to have used the entire wavelength range available while it is clear that the detectors have significant noise on either end, especially the NIR detector. Why include these regions when they are known to only be detector noise?
The concatenation of the data is not clear. Each camera gives a spectrum for each pixel. Pixels are then separated between background and tomatoes. But I am not clear how the spectra are matches between the 2 imagers. When the spectra are concatenated, do they represent the same pixel on the same tomato? This image and spectral processing needs a lot more explanation.
Author Response
Dear reviewer,
We sincerely thank you for reviewing our manuscript and providing valuable comments and suggestions. Your review work plays an extremely important role in the quality and accuracy of our research, and we greatly appreciate your professional guidance. For some questions and suggestions you mentioned, we have made detailed explanations and replies in the attachment. For details, please refer to the attachment. Thanks again for your advice.
Sincerely,
Pan Gao
Shihezi University, Shihezi City, Xinjiang Province, China

Reviewer 2 Report
I found many inaccuracies and formal errors

Author Response
Dear Reviewer:
Thank you very much for reviewing our manuscript and putting forward valuable comments and suggestions. We would like to express our sincere thanks to you for your careful review. In this reply, we respond to your questions and suggestions one by one. Please refer to the attachment for specific reply. Thanks again for your advice.
Sincerely,
Pan Gao
Shihezi University, Shihezi City, Xinjiang Province, China

Round 2
Reviewer 2 Report
The authors have corrected the problems described in the review.
I recommend the corrected manuscript for publication
Author Response
Deer Reviewers,
Thank you very much for your time involved in reviewing the manuscript and your very encouraging comments on the merits. It is a great honor to receive your recognition of this work, and your comments will greatly help to improve the quality of our articles.
Thank you again for your comment on the paper.
Sincerely,
Pan Gao
Shihezi University, Shihezi City, Xinjiang Province, China